# The Evolving Microbiology and Antimicrobial Resistance in Peritonitis of Biliary Origin: An Evidence-Based Update of the Tokyo Guidelines (TG18) for Clinicians

**DOI:** 10.3390/diagnostics15243095

**Published:** 2025-12-05

**Authors:** Elena-Adelina Toma, Octavian Enciu, Gabriela Loredana Popa, Valentin Calu, Dumitru Cătălin Pîrîianu, Andrei Ludovic Poroșnicu, Mircea Ioan Popa

**Affiliations:** 1Carol Davila University of Medicine and Pharmacy, 050474 Bucharest, Romania; elena-adelina.toma@umfcd.ro (E.-A.T.); gabriela.popa@umfcd.ro (G.L.P.); valentin.calu@umfcd.ro (V.C.); catalin.piriianu@umfcd.ro (D.C.P.); andrei.porosnicu@umfcd.ro (A.L.P.); mircea.ioan.popa@umfcd.ro (M.I.P.); 2Surgery Department, Elias University Emergency Hospital, 050474 Bucharest, Romania; 3Parasitic Disease Department, Colentina Clinical Hospital, 050474 Bucharest, Romania; 4The “Cantacuzino” National Medico-Military Institute for Research and Development, 050474 Bucharest, Romania

**Keywords:** biliary peritonitis, cholecystitis, cholangitis, antimicrobial resistance, antimicrobial stewardship, multidrug-resistant organisms

## Abstract

**Background**: Biliary peritonitis is a severe intra-abdominal emergency with high mortality. Effective management requires source control and appropriate antimicrobial therapy. **Methods**: This review synthesizes recent literature (2016–2025), as well as established guidelines recommendations on the evolving microbiology and antimicrobial resistance patterns in biliary tract infections, as data on biliary peritonitis is scarce and relatively heterogeneous. **Results**: The microbiological landscape is stratified by patient history. Community-acquired infections are typically caused by *Escherichia coli*, *Klebsiella pneumoniae*, and *Enterococcus* spp. In contrast, healthcare-associated infections show a shift, with highly resistant pathogens such as *Pseudomonas aeruginosa*, and a tendency towards polymicrobial infections. The rise of multidrug-resistant (MDR) organisms, including extended-spectrum β-lactamase (ESBL)-producing Enterobacterales, Carbapenem-Resistant Enterobacterales (CRE), and Vancomycin-Resistant Enterococci (VRE), is a critical challenge limiting therapeutic options. Resistance patterns vary geographically, necessitating the use of local data. **Conclusions**: This review argues for a paradigm shift from severity-based guidelines to a dual-axis model incorporating resistance risk factors (prior healthcare exposure, previous biliary interventions, a history of MDR infections). We propose a risk-stratified approach to empiric antibiotic selection, emphasizing microbiological diagnostics for therapy de-escalation. Future research should focus on prospective studies, novel antibiotics, and rapid diagnostics.

## 1. Introduction

### 1.1. The Clinical Challenge of Biliary Peritonitis

Biliary peritonitis, the inflammation of the peritoneum resulting from the leakage of bile into the abdominal cavity, represents a surgical emergency associated with substantial morbidity (between 20% and 30%) and a distressingly high mortality rate, which can range from 8% to 50%, especially in elderly patients or those with significant comorbidities [1,2,3]. The underlying etiologies are diverse, most commonly stemming from the perforation of the gallbladder due to severe acute cholecystitis, empyema, or gangrene [4]. Other causes include iatrogenic injury during procedures like cholecystectomy or endoscopic retrograde cholangiopancreatography (ERCP), and, less frequently, spontaneous rupture of the intrahepatic or extrahepatic biliary tree [4,5]. Such ruptures are often precipitated by elevated intraductal pressure from obstructing calculi or strictures [6,7].

The condition rapidly progresses to bacterial peritonitis as microorganisms from the infected biliary system contaminate the peritoneal space. This secondary bacterial infection is the principal driver of sepsis, systemic inflammatory response syndrome (SIRS), and adverse patient outcomes [5,8]. Compounding this issue, bile itself has been demonstrated to impair local host defense mechanisms, thereby creating a more permissive environment for bacterial proliferation and exacerbating the infectious process [9].

### 1.2. The Intersection of Pathophysiology and Microbiology

The clinical progression from a localized biliary tract infection (BTI), such as acute cholecystitis or cholangitis, to diffuse biliary peritonitis represents a pathological continuum [10]. Consequently, the microorganisms responsible for the initial BTI are the very same pathogens that seed the peritoneal cavity upon perforation or leakage [11]. This direct link makes a thorough understanding of the microbiology of cholecystitis and cholangitis fundamental to the effective management of biliary peritonitis. The pathophysiology is rooted in biliary stasis and obstruction, which disrupts the normal antegrade flow of bile and allows for the retrograde ascension of enteric microorganisms from the duodenum [12]. As intraductal pressure rises behind the obstruction, it not only compromises the integrity of the biliary epithelium but also facilitates the translocation of bacteria and their toxins into the systemic circulation and, in the event of rupture, into the peritoneal cavity [2,6,12].

This review operates on the principle that while “biliary peritonitis” is a clinical diagnosis describing a final common pathway, its microbiology is not a distinct entity. Instead, it is a direct reflection of its precursor conditions. The presence of bile salts in the peritoneum causes significant chemical peritonitis, which may impair local defenses and alter membrane permeability differently than in the biliary ducts. Furthermore, the bacterial load in perforation peritonitis is typically higher, and the polymicrobial nature of the infection may be more pronounced due to the sudden release of stagnant, infected bile into a large sterile cavity. Consequently, the pathogen spectrum and resistance behaviors observed in non-perforating BTIs may not be directly applicable to perforation-related peritonitis. Specifically, anaerobic co-infection may be more clinically relevant in peritonitis because of the hypoxic environment within an abscess or collection [12]. The relative scarcity of studies focusing exclusively on the microbiology of biliary peritonitis necessitates logical, evidence-based extrapolation from the far richer body of literature on the microbiology of acute cholecystitis and cholangitis. Still, findings should be interpreted with this distinction in mind. This approach is a limitation but can also be regarded as a necessary methodology for constructing a clinically useful tool.

### 1.3. The Rising Tide of Antimicrobial Resistance

The successful management of biliary peritonitis rests on two interdependent pillars: timely and adequate source control—achieved through surgical debridement, drainage, or percutaneous intervention—and the administration of appropriate antimicrobial therapy [13]. The latter is crucial for controlling sepsis and preventing further complications. However, the efficacy of empiric antibiotic regimens is increasingly jeopardized by the global crisis of antimicrobial resistance (AMR) [14]. Multiple studies have now established that the administration of inadequate initial antibiotic therapy in patients with severe biliary infections is an independent predictor of increased mortality [8,15].

The high mortality associated with biliary peritonitis can be conceptualized as a result of a synergistic, two-sided process. The first aspect is the chemical toxicity of bile, which causes direct tissue damage and, critically, impairs the peritoneum’s local immune defenses [9]. The second attack wave is the subsequent high-inoculum bacterial challenge from the contaminated bile. This combination, in which the host’s innate ability to contain the infection is already compromised, creates an ideal scenario for the development of fulminant sepsis. This dynamic underscores the importance of selecting an appropriate and effective empirical antibiotic from the outset, as the therapeutic window is narrow and the patient’s defenses are weakened.

## 2. Methodology

### 2.1. Study Design and Search Strategy

To ensure a comprehensive synthesis of current evidence, a structured literature search was conducted using the electronic databases PubMed/MEDLINE, Embase, Web of Science, Scopus, and Google Scholar. The search targeted articles published between 1 January 2016, and 31 December 2024. This search was extended to include articles published between 2004 and 2015, considering the scarcity of data regarding biliary peritonitis as a distinct entity. The primary search strategy employed combinations of the following Medical Subject Headings (MeSH) and keywords: “biliary peritonitis,” “acute cholangitis,” “acute cholecystitis,” “bile culture,” “antimicrobial resistance,” “multidrug resistance,” “ESBL,” “CRE,” and “VRE”.

### 2.2. Selection Criteria

The review prioritized peer-reviewed original research articles involving human subjects but also included case reports and limited case series, given the low number of articles specifically addressing biliary peritonitis. Inclusion criteria required that studies provide specific data on microbiological isolates from bile or peritoneal fluid and/or detailed antibiograms. Studies focusing exclusively on pediatric populations or animal models were excluded to maximize the generalizability of findings to adult clinical practice. Articles not available in English were also excluded.

### 2.3. Data Synthesis and Bias Minimization

Retrieved records were screened for relevance by title and abstract. Selected full-text articles were analyzed to extract data regarding pathogen frequency, resistance phenotypes, and genetic resistance mechanisms. Given the scarcity of studies specifically titled “biliary peritonitis,” data were stratified where possible between localized biliary infections (cholecystitis/cholangitis) and those complicated by perforation or bacteremia to infer trends relevant to peritonitis. To minimize selection bias, studies from diverse geographic regions (Europe, Asia, North America) were included to reflect global variations in resistance patterns.

## 3. The Microbiological Landscape of Biliary Tract Infections

The spectrum of pathogens causing BTIs, and by extension biliary peritonitis, is not uniform. It is heavily stratified by the patient’s clinical background, particularly the distinction between community-acquired infections and those associated with healthcare exposure or prior interventions.

### 3.1. The Classic Pathogens in Community-Acquired Biliary Infections

In patients presenting from the community without a recent history of hospitalization or biliary procedures, the responsible pathogens are typically enteric organisms originating from the gastrointestinal tract [16]. Gram-negative bacilli from the *Enterobacteriaceae* family (Order Enterobacterales) have historically been, and remain, the predominant isolates, identified in up to 70% of infections [17,18,19,20,21]. *Escherichia coli* is consistently identified as the most frequent pathogen. Numerous studies report its isolation in 25–40% of positive bile and blood cultures from patients with acute cholangitis or cholecystitis. *Klebsiella pneumoniae* is reliably the second most common Gram-negative organism, with a reported prevalence of 15% to 20% across various studies [22,23,24].

Gram-positive cocci, particularly *Enterococcus species*, are also key pathogens. Once considered by some to be mere colonizers, their pathogenic role is now firmly established, especially in more severe or complicated infections [22,25].

Other, less common but clinically significant isolates in community-acquired BTIs include *Streptococcus* spp., *Enterobacter* spp., *Citrobacter freundii*, *Bacteroides fragilis*, or *Clostridioides* spp. [22].

### 3.2. The Microbial Shift in Healthcare-Associated and Post-Intervention Settings

One of the most critical findings in the recent literature is the dramatic and predictable shift in the microbial flora of patients with healthcare-associated infections (HCAI) or a history of biliary system manipulation. Procedures such as ERCP with sphincterotomy, biliary stent placement, or prior cholecystectomy fundamentally alter the biliary environment, selecting for a different, more resistant cohort of pathogens [25,26].

This microbial evolution is not merely a change in species but reflects a fundamental alteration of the biliary resistome. Interventions like stenting introduce a foreign body that serves as a nidus for biofilm formation, an environment that favors intrinsically more resilient organisms [27]. Subsequent courses of antibiotics then exert powerful selective pressure, eliminating susceptible community flora (such as sensitive *E. coli*) and allowing these more resistant, biofilm-associated organisms to flourish. This process creates a dangerous cycle in which each medical interaction can enrich the biliary system with increasingly challenging bacteria.

In these high-risk patients, there is a significantly higher prevalence of:•*Pseudomonas aeruginosa*: While uncommon in community-acquired BTIs, its prevalence can surge to over 21% in post-cholecystectomy or post-ERCP patients, compared to less than 10% in those without such a history [24,28]. Admission to an intensive care unit (ICU) is another strong risk factor for isolation [29].•*Enterococcus* spp.: The frequency of enterococcal isolation increases markedly in patients with prior interventions. Studies have documented a prevalence of 32.4% in post-cholecystectomy patients and as high as 43.6% in patients with indwelling biliary stents [26,30]. The presence of a biliary drain and the total number of prior interventions are powerful predictors of enterococcal infection.•Other Enterobacterales (*Klebsiella aerogenes*, *Serratia marcescens*, *Citrobacter freundii complex*, *Providencia* spp., *Morganella morganii*): These organisms also show a significantly higher prevalence in patients with a history of cholecystectomy compared to those without previous biliary tract interventions [26,28,30] and are more common in those with biliary reconstructions [31].

Furthermore, polymicrobial infections are far more common in these settings, with some studies reporting that up to 78% of bile cultures grow multiple organisms, particularly in patients with indwelling devices, and that patients with cholangitis have multiple pathogens identified in bile cultures than those with acute cholecystitis [27,29,32]. One study reported that *E. coli* and *E. faecium* were most commonly found in mixed BTI, and these two types of bacteria also showed higher individual rates of AMR [20]. Over 10% of patients had three pathogens identified in their bile cultures, and in 2.39% of cases, four simultaneous bacterial infections were reported. The same study reported a two-fold increase in the proportion of polymicrobial BTI over five years, from 23.08% to 45.67%. Another study reported up to nine distinct microorganisms (including both bacteria and fungi) in one bile culture, with more than half of patients with polymicrobial BTI also having positive blood cultures with at least one bacterial strain present [29]. These findings underline the need for a more aggressive initial treatment in cases where post-interventional infections are likely, including supportive measures even in the absence of clinically evident organ dysfunction. Moreover, microbiological testing of explanted stents could provide important information regarding the current polymicrobial landscape, as multiple pathogens are often involved in biliary stent colonization in patients without cholangitis symptoms [33].

### 3.3. The Role of Anaerobic and Fungal Infections

Anaerobic bacteria, such as *Bacteroides* and *Clostridium* spp., are reported in less than 10% of BTI cases [32,34]. Their presence is particularly suspected in patients with hepatic abscesses, peritonitis, or a history of bilioenteric anastomosis (e.g., choledochojejunostomy), which allows for direct contamination by colonic flora [22,34].

Fungal pathogens, predominantly *Candida* spp., are isolated in less than 5% of community-acquired infections, while some series report a prevalence as high as 55% in patients with previous interventions [21,33,35,36]. Fungal biliary infection is often associated with specific risk factors, including prolonged or broad-spectrum antibiotic therapy, the presence of indwelling biliary stents, and underlying immunosuppression [33,35].

### 3.4. Etiology-Specific Microbiological Variations: Malignant Versus Benign Obstruction

Current evidence is inconclusive regarding distinct microbiological differences between patients with malignant biliary obstruction (MBO) and those with benign causes, such as choledocholithiasis (stones in the common bile duct or gallbladder). In almost half of patients with biliary tract obstruction, regardless of etiology, infection is present, with no association between type of obstruction and specific pathogens, except for one positive significant correlation between *E. coli* and gallstones, as reported by a single-center study [23]. While *E. coli* and *Klebsiella* spp. remain the most common isolates in both groups, pathogens involved in malignant obstruction may exhibit more resistance to cephalosporines or carbapenems in some cohorts [37]. The etiology of biliary tract obstruction is not a certain predictor of AMR, as reported by another study analyzing 568 patients with acute cholangitis [38]. However, malignant obstruction is reportedly a risk factor for higher mortality rates in patients with acute cholangitis [39].

## 4. Evolving Patterns of Antimicrobial Resistance

The selection of an effective empiric antibiotic regimen is complicated not just by the shifting spectrum of pathogens but, more critically, by the escalating rates of antimicrobial resistance among them. This trend threatens the efficacy of nearly all classes of antibiotics used to treat intra-abdominal infections.

### 4.1. The Proliferation of Extended-Spectrum β-Lactamase (ESBL)-Producing Enterobacteriaceae

Extended-spectrum β-lactamases (ESBLs) are enzymes, most commonly found in *E. coli* and *K. pneumoniae*, that hydrolyze and confer resistance to most penicillin and cephalosporin antibiotics, including third- and fourth-generation agents (e.g., ceftriaxone, cefepime) [40]. This renders many first-line empiric therapies for BTIs completely ineffective.

The global prevalence of ESBL-producing bacteria is increasing at an alarming rate in both community and healthcare settings [41,42]. A meta-analysis showed the global intestinal carriage rate in a community setting has increased eight times from the rates observed between 2003–2005 and those reported 2015–2018, while key risk factors for infection with an ESBL-producing organism include prior use of antibiotics (especially cephalosporins and fluoroquinolones), recent hospitalization, and residence in a long-term care facility [40,43]. In the context of biliary infections, the prevalence of ESBL-producing organisms exceeds 23% and increases the risk of cholangitis in patients with biliary obstruction, as concluded by a 2024 study [44]. Another investigation identified an ESBL prevalence of 13.9% in patients with both a prior sphincterotomy and an indwelling stent [33]. The most frequently identified ESBL resistance genes are usually *bla*_CTX-M_ and *bla*_TEM_ in *E. coli*, and *bla*_TEM_ and *bla*_SHV_ for *Klebsiella* spp. [44].

### 4.2. The Emergent Threat of Carbapenem-Resistant Gram-Negative Pathogens

Carbapenem antibiotics, such as meropenem and imipenem, have long served as the last line of defense against severe infections caused by ESBL-producing organisms. Consequently, the emergence and spread of Carbapenem-Resistant Enterobacterales (CRE) represents a grave public health crisis, leaving clinicians with few, if any, effective treatment options [45,46].

While still less common than ESBLs, the prevalence of CRE in biliary infections is rising and warrants significant concern. Recent studies have reported CRE rates ranging from 2.1% to 6.9% in cohorts of patients with cholangitis [44,47]. As with other forms of resistance, the prevalence is significantly higher in patients with a history of biliary stenting, patients with cancer, those with chronic kidney disease, and recent antibiotic treatment [31,47]. *Klebsiella pneumoniae* is the most frequently implicated CRE species in these infections, with the *Klebsiella pneumoniae* carbapenemase (KPC) enzyme being a predominant mechanism of resistance.

In recent years, carbapenem-resistant *Acinetobacter baumannii* has been identified in an increased number of cases, especially in patients who were immunocompromised and required ICU hospitalization. It showed an increased tendency towards multidrug resistance, with over half of the identified strains being carbapenem-resistant, carrying the *bla*_OXA-23_ gene, and requiring treatment with Colistin for effective management [48,49]. However, Colistin-resistant strains of *Acinetobacter baumannii* have been reported in recent years, through gene mutations or gene transfer, which has underlined the need for future development of new treatment strategies [50].

*Pseudomonas aeruginosa* exhibiting resistance to carbapenems has also been reported in BTI patients with severe concomitant acute pancreatitis, with drug resistance rates increasing over the last decade and currently reported at over 60% of cases [49,51].

### 4.3. Resistance in Gram-Positive Pathogens: The Case of Vancomycin-Resistant Enterococci (VRE)

*Enterococcus* species possess a high degree of both intrinsic and acquired antibiotic resistance. A baseline level of resistance is common, with at least one-third of enterococcal isolates from biliary cultures demonstrating resistance to ampicillin, a historically active agent [21].

The primary concern, however, is Vancomycin-Resistant Enterococci (VRE) infections, which have emerged in recent years in patients with acute cholecystitis and/or cholangitis, infections that are particularly problematic in healthcare-associated settings [52]. In patients with BTIs, antibiotic resistance is most often found in *Enterococcus faecium* and *Enterococcus faecalis*, but VRE prevalence can reach almost 90% in certain isolates, such as *Enterococcus casselifavus* [21,36]. VRE strains are rarely encountered in acute cholecystitis patients but have been reported in up to 15% of blood cultures from patients undergoing ERCP after cholecystectomy [28,38,53,54]. A known history of VRE colonization is a significant risk factor for isolating VRE from bile during an acute infection, as are previous invasive procedures, recent antibiotic use, and immunosuppressive therapies, highlighting the importance of reviewing a patient’s past microbiology data [28,55].

Table 1 presents these complex resistance phenotypes in their direct clinical implications, providing a concise reference for clinicians.

## 5. Implications for Antimicrobial Stewardship and Clinical Practice

The evolving microbiological and resistance landscape demands a more nuanced and risk-stratified approach to the management of biliary peritonitis and its precursor infections. A “one-size-fits-all” strategy is no longer viable.

### 5.1. Risk Stratification—The Key to Empiric Therapy

The initial choice of empiric antibiotic therapy should not be based solely on the clinical severity of the presentation. Timely antimicrobial intervention is paramount, because for each 30 min delay in delivering antibiotics, increased mortality by an odds ratio of 1.02, as reported by a prospective study of 356 patients [58].

The rapid but thorough risk assessment for the presence of MDR organisms should be a routine part of the initial patient evaluation. Key risk factors that should prompt consideration of broader-spectrum empiric coverage include [29,36,49]:•Healthcare-Associated Infection: Recent hospitalization (within 90 days), residence in a long-term care facility, or receiving outpatient healthcare (e.g., dialysis).•Prior Biliary Interventions: A history of cholecystectomy, ERCP (especially with sphincterotomy), or the presence of an indwelling biliary stent or drainage catheter.•Recent Antibiotic Use: Exposure to any antibiotic within the preceding 90 days, with particular concern for third-generation cephalosporins and fluoroquinolones.•Known Colonization with MDR Organisms: A documented history of colonization or infection with MRSA, VRE, ESBL-producing organisms, or CRE.•High Comorbidity Burden: A high Charlson Comorbidity Index (e.g., ≥5) has been associated with more resistant pathogens.•Critical Illness: Admission to an ICU setting.

### 5.2. Applying and Adapting Clinical Practice Guidelines

International guidelines, most notably the Tokyo Guidelines (TG18), provide an essential framework for management, but must be applied with flexibility and adapted to local resistance realities [11,59].

-**For Low-Risk Patients:** In a patient presenting from the community with no identified risk factors for MDR organisms, a narrower-spectrum agent may still be appropriate for mild-to-moderate disease. Options with good biliary penetration and safety profiles include a penicillin/β-lactamase inhibitor (usually ampicillin–sulbactam) or a third-generation cephalosporin (especially ceftriaxone, as it exhibits increased levels of biliary excretion) in association with metronidazole in selected cases [60,61]. Fluoroquinolone-based therapy remains an option for patients with known allergies to other antibiotic classes, in association with metronidazole, but with limited efficacy in areas where *E. coli* has over 30% resistance rates in the community [59,61,62].-**For High-Risk Patients:** In any patient with one or more risk factors for MDR pathogens, broader empiric coverage is mandatory, regardless of the initial clinical severity. Piperacillin–tazobactam has traditionally been the workhorse in this scenario, but with resistance rates of Enterobacteriaceae now exceeding 20%, as well as high resistance rates of *Acinetobacter* spp. and *Pseudomonas* spp. in some high-risk populations, its reliability as a single agent is diminishing [63,64]. Cefmetazole has been suggested as an effective alternative for patients with ESBL-producing bacteria, with very good results [56,65]. Patients with severe disease or those with a very high suspicion of an ESBL or CRE infection (known CRE carrier, recent carbapenem use), initial empiric therapy with a carbapenem is warranted pending culture results, but physicians should be aware that meropenem has poorer biliary penetration in obstruction cases, compared to imipenem/cilastatin [59,61,66]. Critically ill patients at risk for *Candida* spp. infection warrants the addition of an antifungal drug, such as fluconazole or echinocandin [11].

The duration of antimicrobial treatments also needs to be adapted to each case, as a key tenet of antimicrobial stewardship is using the shortest effective duration of therapy and modulation of treatment length through close monitoring of C-reactive protein and procalcitonin [67]. A growing body of evidence suggests that for patients with adequately source-controlled biliary infections, shorter courses of antibiotics (3–7 days) are non-inferior to longer courses, and prompt surgical intervention may warrant discontinuation of antibiotherapy after source control, with good outcomes [68,69,70,71]. This practice is crucial for reducing selective pressure, minimizing side effects, and lowering healthcare costs, but close monitoring of outcomes is advisable, as these results of shorter-course therapies have been reported for patients with acute cholangitis or cholecystitis, and not for biliary peritonitis patients [66,70]. According to a review published in 2025 by Sartelli et al., for the Global Alliance for Infections in Surgery, the analysis of multiple trials and randomized controlled trials yielded the same results: in patients with adequate source control, a short course of antibiotics (four days) for patients with mild to moderate disease, while critically ill patients benefit from antimicrobial therapy lasting 7 to 8 days, with no improvement in any outcomes following prolonged therapy [72].

### 5.3. The Critical Role of Microbiological Diagnostics

In an era of rampant resistance, microbiological diagnostics are not an academic exercise; they are a cornerstone of effective and responsible antibiotic use. Obtaining bile and/or blood cultures as soon as possible in all patients with moderate-to-severe BTIs is strongly recommended [13,59]. The primary purpose of these cultures is to facilitate timely de-escalation of therapy or adapt the antimicrobial regimen to individual needs based on resistance profiles. Once a pathogen is identified and its susceptibilities are known, broad-spectrum empiric agents should be narrowed to the most effective, safest, and narrowest-spectrum drug possible [11,32]. This practice preserves broad-spectrum agents for when they are truly needed and minimizes the collateral damage to the patient’s microbiome. While blood cultures have lower sensitivity, a positive result is precise and clinically significant. Bile cultures have a higher diagnostic yield but may sometimes represent colonization rather than true infection, especially in patients with long-term stents [28,62]. Moreover, bile cultures may be positive even in patients who have already received antibiotic therapy and may help select patients for whom antifungal treatment is needed [24]. The concordance between bile and blood cultures is only partial, at less than 40%, and thus both should be collected when feasible [73].

### 5.4. Carbapenem-Sparing Strategies and Future Horizons

The preservation of carbapenems is a global public health priority. Strategies to avoid their overuse are vital. In high-risk patients with suspected resistant Gram-negative infections, newer β-lactam/β-lactamase inhibitor combinations, such as ceftazidime-avibactam, ceftolozane-tazobactam, imipenem-relebactam, or meropenem-vaborbactam, as well as new-generation cephalosporines such as cefiderocol, may have a role in treating confirmed ESBL-producing *Enterobacteriaceae* and certain CRE infections [74,75]. However, clinical data specifically for their use in biliary infections are still limited, and their use should be guided by infectious disease consultation.

The future of BTI management will likely involve the integration of rapid molecular diagnostics. Technologies such as multiplex PCR-based identification can indicate specific pathogens with high sensitivity and specificity, detect specific resistance genes (e.g., *bla*KPC, *bla*NDM, *bla*CTX-M) directly from bile or blood samples within hours rather than days, and could revolutionize empiric therapy, allowing for highly targeted treatment from the outset [76].

Table 2 synthesizes this review into an actionable clinical tool, providing risk-stratified empiric antibiotic recommendations.

## 6. Conclusions and Future Directions

### 6.1. Synthesis of Key Findings

Biliary peritonitis is a life-threatening sequela of common biliary diseases, and its management is increasingly complicated by antimicrobial resistance. This review of recent literature highlights several critical conclusions. First, the microbiology of biliary peritonitis is a direct extension of the primary BTI, and it is profoundly stratified by patient history. The traditional paradigm of community-acquired enteric Gram-negative pathogens is being supplanted in an increasing proportion of patients by a more complex, resistant flora characteristic of healthcare-associated infections, dominated by organisms like *Pseudomonas aeruginosa*, *Enterococcus* spp., and MDR bacteria. Second, resistance to nearly all classes of workhorse antibiotics—including third-generation cephalosporins, fluoroquinolones, and even piperacillin–tazobactam—is high and continues to increase globally. Third, the emergence of ESBL-producing *Enterobacteriaceae* as common pathogens and the rising threat of CRE and VRE in biliary infections pose a profound therapeutic challenge that demands a fundamental shift in clinical strategy. Empiric treatment based on clinical severity alone is no longer sufficient; it must be integrated with a robust assessment of a patient’s risk of harboring a resistant organism.

### 6.2. Identified Gaps in Current Knowledge

Despite the wealth of data on BTIs, several significant knowledge gaps remain, particularly concerning biliary peritonitis. There is a notable lack of prospective, multicenter studies that specifically analyze the microbiology, antimicrobial resistance patterns, and clinical outcomes of biliary peritonitis as a distinct clinical entity. Most current knowledge is extrapolated from studies on cholangitis and cholecystitis. The clinical impact and optimal management of fungal biliary infections, particularly those caused by *Candida* species, are not well defined. Moreover, while newer antibiotic agents with activity against MDR organisms (e.g., novel β-lactam/β-lactamase inhibitor combinations) are available, their efficacy, safety, and optimal place in the treatment of severe, resistant biliary infections require evaluation in dedicated clinical trials.

## Figures and Tables

**Table 1 diagnostics-15-03095-t001:** Key resistance phenotypes and associated antimicrobial non-susceptibility in biliary pathogens.

Resistance Phenotype	Common Pathogens	Key Affected Antibiotics (Drugs Likely Ineffective)	Representative Resistance Rates	Key Citations
ESBL-producing *Enterobacteriaceae*	*E. coli*, *K. pneumoniae*	Penicillins, Cephalosporins (all generations), Aminoglycosides	Prevalence of 15–44% in high-risk BTI cohorts. Carbapenems are the treatment of choice, but resistance is growing.	[23,24,46,56,57]
Carbapenem-Resistant Enterobacterales (CRE) and Gram-negative pathogens	*K. pneumoniae*, *E. coli*, *Acinetobacter baumannii*, *Pseudomonas aeruginosa*	All Penicillins, Cephalosporins, Carbapenems	Prevalence of 4.5–66% in high-risk BTI cohorts. Treatment is extremely difficult, often requiring combination therapy with Colistin and novel agents.	[48,49,51]
Vancomycin-Resistant *Enterococci* (VRE)	*E. faecium* > *E. faecalis*	Vancomycin, often Ampicillin	Prevalence of 8–19% in high-risk BTI cohorts. Treatment options include Linezolid or Daptomycin.	[23,24]

**Table 2 diagnostics-15-03095-t002:** Empiric antibiotic regimens for biliary infections based on severity and resistance risk.

	Mild-to-Moderate Severity (Tokyo Grade I/II)	Severe/Septic Shock (Tokyo Grade III)
**Low Resistance Risk** (Community-acquired, no recent interventions/antibiotics, no recent hospitalization)	**First-line:** Ceftriaxone OR Ampicillin/Sulbactam. **Rationale:** Covers common community pathogens (*E. coli*, *Klebsiella*). **Considerations:** Add Metronidazole if a bilioenteric anastomosis is present. Check local resistance rates for Ceftriaxone [74].	**First-line:** Cefmetazole OR Piperacillin/Tazobactam. **Rationale:** Broader spectrum covering common pathogens plus some *Pseudomonas* and anaerobes. It can also cover ESBL-producing pathogens in areas with high community resistance rates. **Considerations:** Provides more reliable coverage than cephalosporins in sicker patients or in areas with higher ESBL-producers prevalence [56,65].
**High Resistance Risk** (Healthcare-associated, prior biliary intervention, recent antibiotics, known MDR colonization)	**First-line:** Cefmetazole OR Piperacillin/Tazobactam OR Meropenem. **Rationale:** Necessary broad coverage for *Pseudomonas*, *Enterococcus*, and potentially resistant Enterobacterales. **Considerations:** Even with “mild” signs, the risk of a resistant pathogen is high. A carbapenem (Meropenem) may be considered if local piperacillin–tazobactam resistance is high (>15–20%) or if the patient is a known ESBL carrier [11,59,64].	**First-line:** Meropenem OR Imipenem/Cilastatin. **Rationale:** Provides the broadest empiric coverage for ESBL-producers and most other resistant pathogens. **Considerations:** Add Linezolid/Daptomycin if the patient is a known VRE carrier or in a high-prevalence VRE setting. Consult infectious diseases for potential use of novel agents if CRE is highly suspected [11,59,61,64,66].

## Data Availability

No new data were created or analyzed in this study. Data sharing is not applicable to this article.

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
