# Peer review of "The Evolving Microbiology and Antimicrobial Resistance in Peritonitis of Biliary Origin: An Evidence-Based Update of the Tokyo Guidelines (TG18) for Clinicians"

_diagnostics, 2025, doi:10.3390/diagnostics15243095_

Round 1
Reviewer 1 Report
Comments and Suggestions for Authors
This is an interesting paper on the causative and resistant bacteria of bile peritonitis. Please explain whether there are differences between the causative and resistant bacteria in bile peritonitis caused by benign (stones) and malignant (cancer) causes. Please also clearly state the recommended duration of antibiotic administration for bile peritonitis.
Author Response
Dear Reviewer,
We want to express our gratitude for your specific and insightful feedback regarding the role of etiology in biliary infections.
- We agree that distinguishing between malignant and benign obstruction is clinically vital, as the underlying cause significantly influences both the microbiological spectrum and the resistance profile.
In response to your request, we have added a new subsection, "3.3 Etiology-Specific Microbiological Variations: Malignant vs. Benign Obstruction." In this section, we have synthesized recent comparative data regarding this issue.
- We addressed the duration of antibiotic treatment in subsection 5.2., but have added more specific data as indicated by information found in current guidelines, case reports, and reviews regarding this issue.
We believe this information adds value and specificity to the review and provides clearer guidance for clinicians managing these distinct patient populations.
We hope you find the improved version of our manuscript good for publication.
Sincerely,
The authors.
Reviewer 2 Report
Comments and Suggestions for Authors
First of all, thank you very much for this high-quality work. The article is presented as a review synthesizing the literature published between 2016 and 2025; however, the methodology underlying this synthesis is not adequately explained. No search strategy is presented; the databases consulted, search terms, and inclusion or exclusion criteria are not specified, and the study selection process, including the number of records searched or adherence to PRISMA standards, is unclear. Similarly, the authors do not outline any measures taken to minimize bias in evidence selection or interpretation. Furthermore, while the article rightly draws microbiological trends from the broader literature on cholangitis and cholecystitis due to the scarcity of data specific to biliary peritonitis, it should be more clearly acknowledged that this approach is a limitation. The microbiology and resistance patterns in biliary peritonitis may differ from those in non-perforating biliary tract infections because pathogen load, polymicrobial load, and the peritoneal environment can significantly alter the bacterial spectrum and resistance behavior. Consequently, resistance patterns identified in general BTIs may not be directly generalizable to perforation-related peritonitis, and this important distinction needs to be clearly explained and justified in the article.
Author Response
Dear Reviewer,
We sincerely thank you for your thorough review and insightful comments regarding our manuscript. We appreciate your recognition of the work's quality and your constructive feedback on the methodology and scope of the review. We have revised the manuscript to address your concerns as follows:
- Clarification of Review Type and Methodology We agree that the underlying methodology required greater transparency. In response to your comment, we have added a dedicated "Methodology" section (Section 2) to the article.
This article was designed as a narrative review to provide a comprehensive, interpretive synthesis of the evolving microbiological landscape of biliary peritonitis, unlike a systematic review, which answers a narrowly defined question, a narrative approach was selected to allow for a broader exploration of related pathophysiological concepts, genetic mechanisms of resistance, and emerging diagnostic technologies that are not yet standard of care.
To ensure transparency, we have now specified:
- Search Strategy: The databases consulted and the specific search terms used.
- Selection Process: Our inclusion and exclusion criteria
- Bias Minimization: Our strategy to select studies from diverse geographic regions to minimize regional bias in resistance reporting.
- Limitation Regarding Extrapolation of Data We fully accept your point regarding the limitations of extrapolating data from general biliary tract infections to biliary peritonitis. We have expanded the extrapolation process previously mentioned in subsection 1.2. We now detail how the distinct environment of the peritoneal cavity (including the chemical peritonitis caused by bile salts) and the typically higher, polymicrobial pathogen load in perforation scenarios may result in resistance dynamics that differ from those in non-perforating infections. We have emphasized that findings should be interpreted with this information in mind, but, given the available data, we see this as a tool for developing a clinically useful tool, despite the limitation it represents for the article we submitted.
We believe these revisions directly address your concerns and significantly enhance the rigor and clarity of the review.
Sincerely,
Reviewer 3 Report
Comments and Suggestions for Authors
The authors have described the etiology and management in their study entitled 'The Evolving Microbiology and Antimicrobial Resistance in Peritonitis of Biliary Origin: An Evidence-Based Update for Clinicians' however, the description is quite superficial, and the review of literature isn't extensive. This could rather be accepted as a brief report than a review. Authors have not italicized the bacterial names at several places in the manuscript and abstract. Line 116: C. freundii isn't an anaerobe. Polymicrobial infections haven't been discussed in detail. Nomenclature of bacteria and fungi needs correction. Enterobacteriaceae isn't used these days. The review isn't detailed in terms of etiolopathiogensis and management strategies/AMR, and newer references need to be incorporated.
Author Response
Dear reviewer,
Thank you for taking the time to review our manuscript.
While we value your opinion and found your suggestions useful for improving the article, we do not agree with your assessment that it should be considered a brief report rather than a review. This article was intended as a tool for clinicians to approach biliary peritonitis cases in a more efficient way, whether anesthesiologists, surgeons, gastroenterologists, or emergency medicine specialists. There are very few reports regarding intra-abdominal infections of biliary origin, and most of them are old and quite outdated, including international guidelines (which we have included in our references, as they are the only ones available at this time – and might, indeed, be considered older references). As such, we conducted a thorough search of the most relevant literature, while carefully avoiding geographic bias, which could result in a useless tool for many doctors and be far from what we intended.
However, we recognize the value of rigor, appreciate your comments, and have improved the article as stated below.
- We have used italics for bacterial names as indicated. Thank you for pointing this out. We have also changed Enterobacteriaceae to Enterobacterales where it was appropriate.
- freundii is, indeed, a facultative anaerobe bacterium – for ease of reading and in order not to make the article too complicated, we chose to group facultative anaerobes and anaerobes. We amended the phrase to better reflect this delineation.
- We have addressed the issue of polymicrobial infections in the setting of biliary tract infections in a more detailed manner. Thank you for your suggestion. We underlined that most patients with indwelling stents have at least polymicrobial colonization, that the number of pathogens identified in a single bile culture can reach 9 mixed types of bacteria and fungi, and that the most frequent association of pathogens is mentioned.
- We agree that distinguishing between malignant and benign obstruction is clinically vital, as the underlying cause significantly influences both the microbiological spectrum and the resistance profile. In response to your request, we have added a new subsection, "3.3 Etiology-Specific Microbiological Variations: Malignant vs. Benign Obstruction." In this section, we have synthesized recent comparative data on etiology and pathogenesis.
- Management strategies have been discussed in section 5, including the current developments regarding ESBL-producing bacteria and CRE – subsection 5.4. Moreover, we reiterate the purpose of the present article, which is precisely to provide a more accurate assessment tool for patients presenting with a potentially life-threatening ailment, which requires antibiotic treatment, but which may result in initial treatment failure due to either outdated guidelines or local resistance profiles that do not match other geographical areas where data is more widely available.
- Out of the total number of references in our article, over 40 are from the last five years, and out of those, 24 are from 2024/2025, including three from the last month.
- Please kindly observe that the new classification in Orders and Families is still treated differently in different articles. We add some examples from 2016 to 2023 in which freundii appears as anaerobic, as aerobic, or as facultative anaerobic, classified in Enterobacterales (order), or in Enterobacteriaceae (family). As the genetic discussion regarding Escherichia spp. versus Shigella spp., which has been known for many years, there are still debates worldwide.
2016 https://pubmed.ncbi.nlm.nih.gov/27620848/
2017 https://pmc.ncbi.nlm.nih.gov/articles/PMC5853342/
2020 https://pmc.ncbi.nlm.nih.gov/articles/PMC6989070/
2021 https://pmc.ncbi.nlm.nih.gov/articles/PMC9083873/
2023 https://pmc.ncbi.nlm.nih.gov/articles/PMC10135275/
2023 https://pubmed.ncbi.nlm.nih.gov/37055768/
2023 https://pmc.ncbi.nlm.nih.gov/articles/PMC10291173/
Round 2
Reviewer 3 Report
Comments and Suggestions for Authors
The authors have addressed most comments; however, I would like to seek clarification on comments 2 and 7. If you talk about facultative anaerobes, there's a huge list that includes even E. coli, Enterobacter, etc., so it's better to avoid using the term "facultative anaerobe" alongside "anaerobes" and include C. freundii in the initial half of the sentence itself. There are several articles online, but we need to refer to the standard ones.
Author Response
Dear reviewer,
Thank you for taking the time to appraise our manuscript - we are happy that you found the changes we made satisfactory.
You are right about the issue regarding facultative anaerobes - we do think it could be a matter of long and heated debates and this is not what our article is about. We made the changes as suggested.
We also revised some English language aspects.
Thank you again and best regards,
The authors.